# Synthesis of Well-Defined Alternating Copolymer Composed of Ethylmaleimide and Hydroxy-Functionalized Vinyl Ether by RAFT Polymerization and Their Thermoresponsive Properties

**DOI:** 10.3390/polym12102255

**Published:** 2020-10-01

**Authors:** Jin Motoyanagi, Ayaha Oguri, Masahiko Minoda

**Affiliations:** Faculty of Molecular Chemistry and Engineering, Graduate School of Science and Technology, Kyoto Institute of Technology, Matsugasaki, Sakyo-ku, Kyoto 606-8585, Japan; m13615015@edu.kit.ac.jp

**Keywords:** alternating copolymer, RAFT copolymerization, thermoresponsive property, hydroxy-functionalized vinyl ether

## Abstract

Here we report the controlled synthesis of alternating copolymers by reversible addition-fragmentation chain transfer (RAFT) polymerization of hydroxy-functionalized vinyl ether (DEGV) and ethylmaleimide (EtMI) using dithiocarbonate derivative (CPDB) as the RAFT reagent. The resulting alternating copolymer poly[ethylmaleimide-*alt*-(diethylene glycol mono vinyl ether)] (poly(MalMI-*alt*-DEGV)) had a relatively narrow molecular weight distribution (*M*_w_/*M*_n_ < 1.4). These polymers are fully soluble in cold water (5 °C) and an aqueous solution of poly(MalMI-*alt*-DEGV) became turbid upon heating (using an incident wavelength of 600 nm and 1.0 mg mL^−1^ (0.1 wt %) polymer concentration), indicating phase separation above the cloud point temperature (*T*_cp_). The *T*_cp_ of the polymer solution ranged from 15–35 °C, depending on the molecular weight and molecular weight distribution of the polymer.

## 1. Introduction

Recent advances in controlled polymerization techniques have enabled the preparation of various well-defined polymers with predetermined molecular weights and low polydispersity. These controlled polymerization techniques have been used to prepare complex polymeric structures such as diblock, comb-like, and star-shaped constructions, paving the way for new functional materials with applications essential for advanced technologies [1,2,3,4]. However, synthetic polymers are still inferior to natural biological macromolecules in terms of monomer sequence. To address this, many recent studies have reported new methods for the synthesis of polymers with controlled monomer sequences [5,6,7,8,9,10] that show different thermal responses in solution and different phase transitions in the bulk relative to the corresponding random copolymer [11,12,13].

Hydroxy-containing vinyl polymers are well-known synthetic biocompatible materials useful in industrial, commercial, medical, and food applications [14,15,16]. Most hydroxy-containing copolymers obtained by copolymerization with hydrophobic units are lower critical solution temperature (LCST)-type thermoresponsive materials which are soluble in water, but become insoluble above a certain temperature (cloud point temperature; *T*_cp_) upon heating. Such thermoresponsive polymers are of great interest for nanotechnology and biomedical applications [17,18,19,20]. Consequently, controlling the *T*_cp_ by designing the molecular structure of the hydroxy-containing amphiphilic polymer has become an important issue.

Here we report that novel well-defined alternating copolymers of hydroxy-containing vinyl ether monomer (DEGV) and ethylmaleimide (EtMI) obtained by reversible addition-fragmentation chain transfer (RAFT) polymerization exhibit thermoresponsive properties, depending on the molecular weight and polydispersity of the polymer. Quite recently, it has been reported that a series of alternating copolymers consisting of vinyl phenol and *n*-alkyl maleimide showed an upper critical solution temperature (UCST-type) thermal response in aromatic solvents [21], motivating us to synthesize an amphiphilic copolymer which has pendant hydrophilic hydroxy groups and hydrophobic short alkyl chains. We anticipated that this polymer would exhibit LCST-type thermoresponsive properties in aqueous systems. First, we analyzed the polymerization behavior of DEGV and EtMI via RAFT polymerization using dithiocarbonate derivative (CPDB) as the RAFT reagent. In general, functional group-appended poly(vinyl ether)s (polyVEs) are typically synthesized by cationic polymerization rather than radical polymerization due to their low radical reactivity, but this approach requires protection and deprotection steps to produce polyVEs with pendant polar functional groups such as OH or CO_2_H. The synthesis of hydroxy-containing vinyl ether polymers by RAFT homopolymerization without the use of protecting groups has been extensively investigated by Sugihara to produce various hydroxy-containing polyVEs [22]. However, the synthesis of alternating hydroxy-containing copolymers has not been reported to date. Here, we investigated the RAFT copolymerization of DEGV and EtMI to synthesize alternating copolymers (Scheme 1) comprising a hydroxy-containing unit positioned adjacent to an EtMI unit. In addition, the thermo-responsive properties of the obtained copolymers were examined by changes in the solubility of aqueous solutions as a function of temperature.

## 2. Materials and Methods

### 2.1. Chemicals and Reagents

Unless otherwise stated, all commercial reagents were used as received. *N*-Ethylmaleimide (EtMI; Tokyo Chemical Industry Co., Ltd., Tokyo, Japan, 98%) and 2,2′-azobis(isobutyronitrile) (AIBN; FUJIFILM Wako Pure Chemical Corporation, Osaka, Japan, 98%) were used as received. Diethylene glycol mono vinyl ether (DEGV; Maruzen Petrochemical Co., Ltd., Tokyo, Japan, 99.4%) was dried overnight over KOH pellets, and distilled twice over CaH_2_. 2-Cyano-2-propyl benzodithioate (CPDB) was prepared according to the literature [23].

### 2.2. Methods

^1^H and ^13^C NMR spectra were recorded at 25 °C on a Bruker model AC-500 spectrometer (Bruker, Billerica, MA, USA), operating at 500 and 125 MHz, respectively, where chemical shifts (*δ* in ppm) were determined with respect to non-deuterated solvent residues as internal standards. Analytical size exclusion chromatography (SEC) was performed in tetrahydrofuran (THF) at 40 °C, using 8.0 mm × 300 mm gel columns (Shodex KF-804 × 2) on a TOSOH model DP-8020 (TOSOH, Tokyo, Japan) with an RI-8022 RI detector (TOSOH, Tokyo, Japan). The number-average molecular weight (*M*_n_) and polydispersity ratio (*M*_w_/*M*_n_) were calculated from the chromatographs with respect to 15 polystyrene standards (Scientific Polymer Products, Inc., Ontario, NY, USA; *M*_n_ = 580–670,000, *M*_w_/*M*_n_ = 1.01–1.07). Elemental analyses were recorded on a Yanaco CHN CORDER MT-5 instrument (Yanaco, Tokyo, Japan). UV-vis spectra were recorded using a quartz cell of 1 cm path length on a SHIMADZU Type UV-2550 spectrometer (SHIMADZU, Kyoto, Japan).

### 2.3. Copolymerization of DEGV and EtMI under RAFT Polymerization Conditions

RAFT copolymerization of DEGV and EtMI was carried out with CPDB as a chain transfer agent and AIBN as an initiator. To a solution of DEGV (100 mg, 760 μmol), EtMI (95 mg, 760 μmol), and CPDB (3.4 mg, 15 μmol) in 1,2-dichloroethane (770 mg) was added AIBN (2.4 mg, 15 μmol) in a glass tube ([DEGV]_0_/[EtMI]_0_/[AIBN]_0_/[CPDB]_0_ = 50/50/1/1). The resulting solution was degassed by three freeze-pump-thaw cycles, and then the glass tube was sealed under vacuum and heated at 60 °C for 1–8 h and quenched by rapid cooling. The reaction mixture was analyzed by SEC and ^1^H NMR spectroscopy. The THF solution of reaction mixture was poured into a large amount of hexane to precipitate the polymers to remove the unreacted monomers. The resultant polymer was collected by centrifugation and dried under reduced pressure. The isolated polymer structure was analyzed by ^1^H NMR measurement.

## 3. Results and Discussion

### 3.1. Copolymerization of DEGV and EtMI under RAFT Polymerization Conditions

First, DEGV and EtMI were copolymerized using AIBN with or without CPDB (a chain transfer agent widely used in RAFT polymerization [23,24,25,26]) as the RAFT agent. The reaction was conducted in 1,2-dichloroethane at 60 °C ([DEGV]_0_/[EtMI]_0_/[AIBN]_0_/[CPDB]_0_ = 50/50/1/1 or 50/50/1/0, [DEGV]_0_ + [EtMI]_0_ = 20 wt %). SEC curves of the obtained polymers with or without CPDB were unimodal, but their molecular weight (MW) and molecular weight distribution (MWD) profiles were clearly different: polymers obtained with CPDB had lower MW and narrower MWD than polymers obtained without CPDB (polymer (with CPDB): *M*_n_ = 7100, *M*_w_/*M*_n_ = 1.38; polymer (without CPDB): *M*_n_ = 47,000, *M*_w_/*M*_n_ = 2.91) (Figure 1). Since the average molecular weight of the polymer product is reduced by chain transfer reactions [27,28], CPDB apparently acts as a chain transfer agent in this copolymerization system. Next, we investigated the RAFT copolymerization behavior in detail. As shown in Figure 2b, consumption of the two vinyl groups was individually followed using ^1^H NMR spectroscopy and smoothly increased at almost the same rate to around 80% within 8 h. SEC traces of the obtained copolymers were unimodal and shifted to a higher molecular weight region as polymerization proceeded while maintaining *M*_w_/*M*_n_ values of ca. 1.4 or below (Figure 2a,c). The *M*_n_ of the resultant copolymers increased in direct proportion to the monomer conversion and was in good agreement with the calculated values (Figure 2c). These results suggested that the RAFT copolymerization strategy used in this study could provide an alternating sequence of DEGV and EtMI in a controlled manner. Poly(DEGV-*co*-EtMI) was purified by subjecting the reaction mixture to reprecipitation from THF into hexane to remove unreacted monomers, followed by centrifugation. Figure 3 shows the ^1^H NMR spectrum of the isolated copolymer (conversion of the two vinyl groups = 78%, *M*_n_ = 8500; *M*_w_/*M*_n_ = 1.41). In addition to the signals from the broad aliphatic protons arising from repeating DEGV and EtMI units at *δ* 0.9–4.5 ppm, the aromatic protons were clearly observed at *δ* 7.4, 7.6, and 7.8 ppm and were assignable to the dithiobenzoate moiety at the *w*-terminus. The integration ratio for the characteristic signals (peak d and peaks e,h,j–m in Figure 3) supported the formation of a copolymer comprising DEGV and EtMI with a 1:1 composition ratio. The RAFT copolymerization results for DEGV:EtMI = 1:1 are summarized in Table 1. All the obtained copolymers had similar composition ratios of DEGV/EtMI ≈ 45/55, irrespective of the monomer conversion. Using elemental analysis, the composition ratios were determined by comparing the nitrogen and carbon atom content since only the EtMI unit contains nitrogen atoms (Table 1). The composition ratios determined by ^1^H NMR and elemental analysis and the conversion ratios of the monomers are in good agreement with each other. ^13^C NMR spectroscopy was used to verify the sequence of the DEGV and EtMI units in the obtained copolymer (Figure 4). The signals from the methyl carbon and the carbonyl carbon of the EtMI unit were compared with those for the EtMI homopolymer (polyEtMI) [29,30]. The methyl carbon signals from the EtMI units in the obtained copolymer appeared at around *δ* 12.7 ppm (peak 1), in sharp contrast to polyEtMI, where they appeared around *δ* 12.5 ppm (peak 1’). In addition, the signals at *δ* 174–180 ppm assigned to the carbonyl carbon in the obtained copolymer were quite different from those of polyEtMI. Given the polymerization behavior indicating similar consumption of the two co-monomers, these NMR data support an alternating sequence of DEGV and EtMI.

### 3.2. Thermoresponsive Behavior of Poly(DEGV-alt-EtMI) in an Aqueous Medium

The resultant poly(DEGV-*alt*-EtMI) was poorly soluble in water at room temperature but dissolved to give a clear solution on cooling to 0 °C, demonstrating the possibility of an LCST-type thermoresponsive polymer. We determined the *T*_cp_ values for the polymer by investigating the thermoresponsive behavior of aqueous solutions upon heating. Furthermore, since poly(DEGV-*alt*-EtMI) was synthesized by RAFT polymerization, we could easily prepare polymers with different polymer chain lengths. Using the alternating copolymers shown in Table 2 (entries 1–3), we also investigated the dependence of the polymer chain length on the temperature response. The temperature of an aqueous solution of poly(DEGV-*alt*-EtMI) sample (0.1 wt %) was gradually increased at a constant rate and the optical transparency of the solution was monitored by the change in transmittance at 600 nm (Figure 5a). Poly(DEGV-*alt*-EtMI) comprising a longer polymer chain (*M*_n_ = 8500) showed a sharp decrease in transmittance and a *T*_cp_ (transmittance of 50%) of about 18 °C. In contrast, a polymer comprising shorter polymer chains (*M*_n_ = 3300) did not cause a sharp drop in transmittance but rather a gradual decrease, resulting in a higher *T*_cp_ of 33 °C. These results suggest that low molecular weight polymers show a phase transition temperature dependent on the molecular weight, whereas polymers with molecular weights above a certain level do not change the phase transition temperature. Next, we investigated the relationship between molecular weight distribution (MWD) and thermoresponsive behavior. We prepared poly(DEGV-*alt*-EtMI) with a wider MWD than the poly(DEGV-*alt*-EtMI) obtained by RAFT polymerization by copolymerizing DEGV and EtMI under conventional radical polymerization conditions with 1-dodecanethiol as a chain transfer agent (Table 2, entries 4 and 5). Interestingly, the transmittance changes of aqueous solutions of these copolymers upon heating were similar and not dependent on molecular weight, in contrast to copolymers with a narrow MWD (Figure 5b). Furthermore, comparison of two polymer solutions with similar molecular weights but different MWDs showed that the polymer with a wide MWD exhibited a sharp change in transmittance with temperature and the phase transition temperature decreased by ca. 20 degrees. This was as expected: a copolymer with a wide MWD contains a large amount of relatively high molecular weight copolymer which becomes hydrophobic at lower temperature [31,32], resulting in a lower phase transition temperature. These results suggest that the phase transition temperature can be controlled by controlling the molecular weight and the MWD by RAFT polymerization.

## 4. Conclusions

We succeeded in the synthesis of new alternating copolymers [poly(DEGV-*alt*-EtMI)] by RAFT copolymerization of a hydroxy appended VE (DEGV) and EtMI. Aqueous solutions of poly(DEGV-*alt*-EtMI)s showed thermoresponsive behavior, became cloudy at temperatures above *T*_cp_, and *T*_cp_ was closely related to the molecular weights and molecular weight distributions of poly(DEGV-*alt*-EtMI).

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
