# Peer review of "Synthesis of Well-Defined Alternating Copolymer Composed of Ethylmaleimide and Hydroxy-Functionalized Vinyl Ether by RAFT Polymerization and Their Thermoresponsive Properties"

_polymers, 2020, doi:10.3390/polym12102255_

Round 1

Reviewer 1 Report

Journal: Polymers

Manuscript ID: polymers-932059

Title: Synthesis of well-defined alternating copolymer composed of ethylmaleimide and hydroxy-functionalized vinyl ether by RAFT polymerization and there thermoresponsive properties

This manuscript describes the controlled synthesis of alternative copolymers by RAFT polymerization of hydroxy-functionalized vinyl ether (DEGV) and ethylmaleimide (EtMI), and studies the lower critical solution temperature (LCST)-type thermoresponsive property upon heating. In general, this work is well conducted. Therefore, I recommend the paper for publication in Polymers after revision. My comments are as follows.

Comments:

  1. In introduction section, what are the advantages of this research compared with the reported hydroxy-containing vinyl ether polymers listed by the authors?
  2. I suggest that the authors should put size exclusion chromatography (SEC)about polymers with or without CPDB in the article, which is more intuitive. In Figure 2, which groups does the peak of δ5-2.4ppm belong to?
  3. It can be seen from Figure 1 and Table 1 that the Mw/Mnvalue increases with the increase of reaction time, why? If the reaction time is greater than 8 h, will the Mw/Mn value increase?
  4. In“1. Copolymerization of DEGV and EtMI under RAFT polymerization conditions” section, the authors describe that “In the elemental analysis of the copolymers, the composition ratios were determined by comparing the contents between nitrogen and carbon atoms”, however, element analysis was not mentioned in “2.2. Methods” section.
  5. In Table 1, theconversion of DEGV and EtMI were 75% and 79%, respectively, but in the description of the authors, the conversion of DEGV and EtMI were reached to 80% within 8 h.
  6. The abbreviation that appears in this article needs to be unified. Andthe abbreviation that appears for the first time needs to be written in its full name. For example, “poly(MalMI-alt-DEGV)” in a
  7. There are someformatting errors and grammatical errors in this paper. For example, in Page 7 Line 226, “ Commun. 2020, 56,” should be corrected as “Chem. Commun. 2020, 56,”. In Page 3 Line 105, “The Mn of the resultant copolymers were increased in direct proportion” needs to be modified. There are other similar mistakes, which aren’t listed here, please revise after checking carefully.

Author Response

We appreciate your valuable comments and suggestions. With careful consideration of your comments, we have modified the manuscript and answered to the questions as follows.

  1. In introduction section, what are the advantages of this research compared with the reported hydroxy-containing vinyl ether polymers listed by the authors?

--> Considering your comment, in introduction section, we have added the sentence as follows:

      “Quite recently, it has been reported that a series of alternating copolymers consisting of vinyl phenol and n-alkyl maleimide showed an upper critical solution temperature (UCST-type) thermal response in aromatic solvents [6], motivating us to synthesize an amphiphilic copolymer having pendant hydrophilic hydroxy groups and hydrophobic short alkyl chains. We anticipated that this polymer would exhibit LCST-type thermoresponsive properties in aqueous systems.”

  1. I suggest that the authors should put size exclusion chromatography (SEC) about polymers with or without CPDB in the article, which is more intuitive. In Figure 2, which groups does the peak of δ1.5-2.4ppm belong to?

--> Considering your comment, we have added the SEC curves of polymers obtained with or without CPDB as Figure 1.  And, in Figure 2, the peaks at δ1.5-2.4ppm could be characterized to main chain of poly(DEGV-alt-EtMI). For better understanding, we modified original manuscript, and we also revised Figure 2 where the main chain are also characterized.

      “In addition to the signals from the broad aliphatic protons arising from repeating DEGV and EtMI units at δ 0.9−4.5 ppm,”

  1. It can be seen from Figure 1 and Table 1 that the Mw/Mn value increases with the increase of reaction time, why? If the reaction time is greater than 8 h, will the Mw/Mn value increase?

-->As you commented, the Mw/Mn value increases as the reaction time increases. Recently, it has been reported that the controlled radical homopolymerization of vinyl ethers with a hydroxy group [Sugihara, S.; Yoshida, A.; Kono, T.; Takayama, T.; Maeda, Y. “Controlled Radical Homopolymerization of Representative Cationically Polymerizable Vinyl Ethers”, J. Am. Chem. Soc. 2019, 141, 13954−13961]. In this journal, the Mw/Mn value of obtained polymers increased as the reaction time increases. To our knowledge, it has widely been recognized that hydroxy group causes chain transfer reaction under radical polymerization conditions. Therefore, it is considered that the chain transfer reaction also occurred in this polymerization reaction and the Mw/Mn value increased at the end of polymerization.

      When the reaction time was extended to 24 h, the monomer conversion did not increase, and the Mw/Mn value maintained the same value.

  1. In “3.1. Copolymerization of DEGV and EtMI under RAFT polymerization conditions” section, the authors describe that “In the elemental analysis of the copolymers, the composition ratios were determined by comparing the contents between nitrogen and carbon atoms”, however, element analysis was not mentioned in “2.2. Methods” section.

--> We have added the elemental analysis equipment in “2.2 Methods” section as follows:

“Elemental analyses were recorded on a Yanaco CHN CORDER MT-5 instrument.”

  1. In Table 1, the conversion of DEGV and EtMI were 75% and 79%, respectively, but in the description of the authors, the conversion of DEGV and EtMI were reached to 80% within 8 h.

--> According to your suggestion, we have revised the sentence as follows:

“… were smoothly increased at almost the same rate to around 80% within 8 h.”

  1. The abbreviation that appears in this article needs to be unified. And the abbreviation that appears for the first time needs to be written in its full name. For example, “poly(MalMI-alt-DEGV)” in abstract.

--> We added the full name of poly(MalMI-alt-DEGV) as poly[ethylmaleimide-alt-(diethylene glycol mono vinyl ether)].

  1. There are some formatting errors and grammatical errors in this paper. For example, in Page 7 Line 226, “Chem. Commun. 2020, 56,” should be corrected as “Chem. Commun. 2020, 56,”. In Page 3 Line 105, “The Mn of the resultant copolymers were increased in direct proportion” needs to be modified. There are other similar mistakes, which aren’t listed here, please revise after checking carefully.

--> We scrutinized format and grammatical mistakes and asked a native person to proofread our manuscript.

Reviewer 2 Report

please see attached file.

Author Response

We appreciate your valuable comments and suggestions. With careful consideration of your comments, we have modified the manuscript and answered to the questions as follows.

  1. Title – spelling – “Synthesis of well-defined alternating copolymer composed of ethylmaleimide and hydroxy-functionalized vinyl ether by RAFT polymerization and their thermoresponsive properties” “there” should be replaced by “their”.

--> We modified the title and asked a native person to proofread our manuscript.

  1. Abstract does not describe fully the key results - LCST - what was the range of temperatures, conditions?

--> According to your comment, we have modified the introduction part.

      “These polymers are fully soluble in cold water (5 °C), and an aqueous solution of poly(MalMI-alt-DEGV) became turbid upon heating (using an incident wavelength of 600 nm and 1.0 mg mL−1 (0.1 wt%) polymer concentration), indicating phase separation above the cloud point temperature (Tcp). The Tcp of the polymer solution ranged from were 15−35 °C, depending on the molecular weight and molecular weight distribution of the polymers.”

  1. Page 1, introduction, lines 24-25: “Recently advances in controlled polymerization techniques have enabled preparation of various well-defined polymers with predetermined molecular weights and narrow polydispersity”. Polydispersities are low, molecular weight distributions are narrow.

--> According to your comment, we have modified the sentence as follows:

      “Recent advances in controlled polymerization techniques have enabled the preparation of various well-defined polymers with predetermined molecular weights and low polydispersity.”

  1. Page 2: lines 42-43: Why were these specific monomers chosen? Why did you need an alternating microstructure? It was stated in lines 52-52 that hydroxy-containing polymers with alternating microstructure have not been reported. There must be a strong basis for choosing hydroxy-functional vinyl ethers and combining them with this particular maleimide.

--> Considering your comment, we have added the sentence as follows:

      “Quite recently, it has been reported that a series of alternating copolymers consisting of vinyl phenol and n-alkyl maleimide showed an upper critical solution temperature (UCST-type) thermal response in aromatic solvents [6], motivating us to synthesize an amphiphilic copolymer having pendant hydrophilic hydroxy groups and hydrophobic short alkyl chains. We anticipated that this polymer would exhibit LCST-type thermoresponsive properties in aqueous systems.”

  1. Page 3, lines 96-99: “The polymers obtained with CPDB showed lower MW and narrower MWD than those” The polymers obtained with CPDB showed lower MW and narrower MWD than those obtained without CPDB…” What was the reason for the lower MW? Were the formulations different; did they use the same concentration of AIBN?

--> To our knowledge, it has widely been recognized that, in radical polymerization, the addition of a chain transfer agent promotes the chain transfer reaction and reduces the molecular weight of the produced polymer. (Macromolecules 1999, 32, 6019–6030). We have added the sentences and references as follows;

      “Since the average molecular weight of the polymer product is reduced by chain transfer reactions [9], …”

      As ref. 9 (a) Rudin, A.; Choi, P. The Elements of Polymer Science & Engineering, 3rd ed.; Elsevier: New York, NY, USA, 2013; pp. 341–389. (b) Heuts, J. P. A.; Davis, T. P.; Russell, G. T. Comparison of the Mayo and Chain Length Distribution Procedures for the Measurement of Chain Transfer Constants. Macromolecules 1999, 32, 6019–6030.

  1. Page 4, Figure 3: Can the maleimide be homopolymerized? Are there additional references for this? Maleimides and anhydrides almost always need to be copolymerized with another monomer.

--> The maleimide can be homopolymerized smoothly. Considering your comment, we have added the references as follows:

      As ref. 10 (a) Matsuoto, A.; Kubota, T.; Otsu, T. Radical polymerization of N-(alkyl-substituted phenyl)maleimides: synthesis of thermally stable polymers soluble in nonpolar solvents. Macromolecules 1990, 23, 4508–4513. (b) Nakayama, Y.; Smets, G. Radical and anionic homopolymerization of maleimide and N-n-butylmaleimide. J. Polym. Sci. A Polym. Chem. 1967, 5, 1619–1633.

  1. Page 5, Section 3.2: “LCST” was misspelled in a few places as “LSCT”. You observed an LCST that was tunable with chain length. However, did you observe the LCST as a function of heating rate, concentration, too? Based on the description here, only really cloud points were measured and true LCSTs were not ascertained. An LCST is really a concentration-dependent quantity – you need to measure solutions with various concentrations.

--> Thank you for your valuable comments and suggestions. As you pointed out, we also recognize that further experiments are necessary for clear the thermoresponsive behavior of our copolymers. We are going to investigate the thermoresponsive behavior of our copolymers by DCS and 1H NMR. Thus, in this journal, we only highlight the cloud point temperature of the polymer solutions.

Reviewer 3 Report

The authors investigated the synthesis of a copolymer derived from diethylene glycol mono vinyl ether (DEGV) and N-ethylmaleimide (EtMI) as well as the thermoresponsive behaviour of the obtained product in aqueous solutions. The polymerisation process was performed as a conventional free radical polymerisation with 2,2'-azobis(isobutyronitrile) as initiator and also as a controlled process in the presence of a RAFT agent: 2-cyano-2-propyl benzodithioate. Evidences were provided that the obtained product is a alternating copolymer, as well as it exhibit LCST.

The novelty of the investigation is the synthesis of a new alternating copolymer of hydroxy-containing vinyl ether monomer without applying protection/ deprotection techniques. Further dependences of LCST on molecular weight and molecular weight distribution were also studied.

The manuscript is well written however there are some issues that should be addressed in order to recommend publication.

1. The Introduction section could include brief information on the essential properties (reactivity ratios) of the monomers yielding alternating copolymers which is related to the choice of the comonomers used in the study.

2. Lines 50-52: Please provide reference in “The synthesis of hydroxy-containing vinyl ether polymers by RAFT homopolymerization without protection group has been extensively investigated by Sugihara to produce various hydroxy-containing polyVEs.”

3. Lines 157-159: The sentence “These results suggest that low molecular weight polymers show a phase transition temperature dependent on the molecular weight, whereas the molecular weight above a certain level does not change the phase transition temperature.” - needs more precise wording.

4. The authors observed that the copolymer with a broad MWD did not display dependence of LCST on the molecular weight which was explained with the presence of high molecular weight fraction. It could be an observation for a given concentration of the copolymer solution. Therefore, it is advisable  measurements at different copolymer concentrations to be performed.

5. Line 178: “3Determined by elemental analysis.” to be deleted.

6. The Conclusion section is too brief and general. It should be extended with the important and specific results obtained in the investigation.

Author Response

We appreciate your valuable comments and suggestions. With careful consideration of your comments, we have modified the manuscript and answered to the questions as follows.

  1. The Introduction section could include brief information on the essential properties (reactivity ratios) of the monomers yielding alternating copolymers which is related to the choice of the comonomers used in the study.

--> Considering your comment, we have added the sentence as follows:

      “To achieve the alternating structure, we focus on the copolymerization of an electron-rich and an electron-deficient vinyl monomers, which have been widely recognized to afford alternating copolymers via radical polymerization mechanism [2a].”

  1. Lines 50-52: Please provide reference in “The synthesis of hydroxy-containing vinyl ether polymers by RAFT homopolymerization without protection group has been extensively investigated by Sugihara to produce various hydroxy-containing polyVEs.”

--> Considering your comment, we have added the references as follows:

      As ref.7.      Sugihara, S.; Yoshida, A.; Kono, T.; Takayama, T.; Maeda, Y. Controlled Radical Homopolymerization of Representative Cationically Polymerizable Vinyl Ethers. J. Am. Chem. Soc. 2019, 141, 13954−13961.

  1. Lines 157-159: The sentence “These results suggest that low molecular weight polymers show a phase transition temperature dependent on the molecular weight, whereas the molecular weight above a certain level does not change the phase transition temperature.” - needs more precise wording.

--> According to your comment, we have modified the sentence as follows:

      “These results suggest that low molecular weight polymers show a phase transition temperature dependent on the molecular weight, whereas polymers with molecular weights above a certain level do not change the phase transition temperature.”

  1. The authors observed that the copolymer with a broad MWD did not display dependence of LCST on the molecular weight which was explained with the presence of high molecular weight fraction. It could be an observation for a given concentration of the copolymer solution. Therefore, it is advisable  measurements at different copolymer concentrations to be performed.

--> Thank you for your valuable comments and suggestions. As you pointed out, we also recognize that further experiments are necessary for clear the thermoresponsive behavior of our copolymers. We are going to investigate the thermoresponsive behavior of our copolymers by DCS and 1H NMR.

  1. Line 178: “3Determined by elemental analysis.” to be deleted.

--> We deleted the sentence.

  1. The Conclusion section is too brief and general. It should be extended with the important and specific results obtained in the investigation.

--> According to your comment, we have modified the conclusion section as follows:

      “We succeeded in the controlled synthesis of new alternating copolymers [poly(DEGV-alt-EtMI)] by RAFT copolymerization of a hydroxy appended VE (DEGV) and EtMI. As results of the copolymerization behavior and the structural analysis of the copolymers, the resulting copolymer had alternating structure and a relatively narrow molecular weight distribution (Mw/Mn < 1.4). The aqueous solutions of poly(DEGV-alt-EtMI)s showed thermoresponsive behavior, became cloudy at temperatures above Tcp, and the Tcp was closely related to the molecular weights and molecular weight distributions of poly(DEGV-alt-EtMI).”

Round 2

Reviewer 1 Report

Journal: Polymers Manuscript ID: polymers-932059 Title: Synthesis of well-defined alternating copolymer composed of ethylmaleimide and hydroxy-functionalized vinyl ether by RAFT polymerization and there thermoresponsive properties This manuscript describes the controlled synthesis of alternative copolymers by RAFT polymerization of hydroxy-functionalized vinyl ether (DEGV) and ethylmaleimide (EtMI), and studies the lower critical solution temperature (LCST)-type thermoresponsive property upon heating. I have carefully examined the revised manuscript. The authors have revised the manuscript according to my comments, and I recommend the manuscript for publication in Polymers.

Reviewer 2 Report

I am satisfied with the changes made.